# Risk of infection in neonates born in accidental out-of-hospital deliveries

**Chia-Jung Chang**[1]**, Hsin Chi**[1,2,3]**, Wai-Tim Jim**[1,2,3]**, Nan-Chang Chiu**[1,2,3]**, Lung Chang**[1,2,3]*

1 Department of Pediatrics, MacKay Children's Hospital and MacKay Memorial Hospital, Taipei, Taiwan, 2 Department of Medicine, MacKay Medicine College, New Taipei City, Taiwan, 3 MacKay Junior College of Medicine, Nursing and Management, Taipei, Taiwan

* covis77@yahoo.com.tw

**Data Availability Statement:** All relevant data are within the paper and its Supporting Information files.

## Abstract

Accidental out-of-hospital deliveries (OHDs) are known to have a higher incidence of maternal and neonatal complications. However, neonatal infection related to OHDs has not been studied. The aim of this study was to determine the infection risk of OHDs. This retrospective cohort study enrolled neonates admitted at a children's hospital in an urban setting from January 2004 to December 2017. Accidental OHDs were compared with in-hospital births, and neonatal infection was assessed. This study also investigated both maternal and neonatal risk factors associated with OHDs. A cohort of 158 OHD neonates was enrolled, of whom 29 (23.2%) were preterm. Prematurity and low birth weight were significantly associated with OHD. Eight neonates in the OHD cohort had a documented infection within the first 72 hours of life, which was 11-fold higher than infections documented for the in-hospital births. Multivariate analysis identified low birth weight as the only factor independently associated with increased risk of infection in OHD neonates. Several specific characteristics of mothers with OHDs were identified. Forty-nine (31%) OHD mothers lacked antenatal care, and 10 (6.3%) were unaware of their pregnancies. The OHD group comprised of more teenage mothers compared to the in-hospital deliveries category. Neonatal infection was more prevalent among OHDs than for in-hospital deliveries, and the infection rate was associated with low birth weight. Hospitalization for further care and observation is suggested for the OHD neonates. Social support should be provided for populations with an increased risk of OHD, such as teenage mothers.

## Introduction

Accidental out-of-hospital deliveries (OHDs) constitute <1% of all live births in most developed countries [1–8]. These emergency births differ from planned home births and in-hospital births because OHDs usually happen accidentally at home or en route to the hospital [3, 4]. These neonates might be delivered in relatively harsh conditions, and an increase in adverse outcomes has been reported for both mothers and neonates involved in OHD. Neonates born in an unplanned setting have high rates of respiratory distress, hypoglycemia, and

**Funding:** The authors received no specific funding for this work.

**Competing interests:** The authors have declared that no competing interests exist.

hypothermia, contributing to a considerably greater chance of requiring admission to special care nurseries or intensive care units (ICUs) than do neonates from similar in-hospital births [1–3, 7, 8]. In comparison with in-hospital deliveries, OHDs are associated with a higher rate of maternal complications, including extensive lacerations of the birth canal, uterine rupture, and post-partum hemorrhage [9–12]. Khupakonke et al. also demonstrated LBW to be a predicting factor of OHD [11]. Since the maternal and neonatal outcomes of OHDs are substantially different from those of in-hospital births, studying the risk factors in this specific group of patients is worthwhile.

The unpredictable characteristics of OHDs mean that neonates are born in inappropriate locations without midwives or medical professionals on standby [1, 2, 4]. These adverse circumstances and poor perinatal care, such as being born in contaminated places and suboptimal cord practices, may increase the risk of infection from OHDs, which can also result in neonatal sepsis. Neonatal infection is the most common cause of neonatal mortality in developing countries [13, 14]. However, previous studies of OHDs have not clearly proposed the risk factors associated with infection among these neonates. Therefore, the aim of this study was to investigate the risk of infection in neonates born outside hospitals. Furthermore, the maternal characteristics associated with OHDs can be identified antenatally.

## Materials and methods

### Design and setting

This retrospective cohort study was conducted at MacKay Children's Hospital, an urban setting in the capital of Taiwan. We enrolled all neonates with birth records in our hospital from January 2004 to December 2017. Stillbirths were excluded. This study was approved by the Institutional Review Board of MacKay Children's Hospital (IRB No. 18MMHIS112). The information we obtained were lists of fully anonymized data without chart number or patient's name.

### Case definition

In this study, the inclusion criteria for OHDs were neonates who were born accidentally in places other than hospitals and without assistance from health care providers. In-hospital deliveries were defined as neonates born in the delivery room, operation room, or other health facilities. To evaluate the prevalence and compare the variables of OHDs, the obstetrics records and neonatal data during the study period were retrieved from the medical record database of MacKay Children's Hospital. Fig 1 presents the flowchart of patient enrollment.

### Study variables

Gestational age (GA), gender, birth weight, duration of hospitalization, ICU stay, maternal medical history, place of labor, laboratory tests, and microbiological studies were reviewed. We recorded the duration of antimicrobial therapy in the first 72 hours of neonatal life for early-onset infections. Cutting of the umbilical cord with unsterilized scissors was considered a suboptimal cord practice. Prematurity was defined as a GA of <37 weeks. Low birth weight (LBW) was defined as weight at birth of <2500g. Teenage mothers were regarded as those aged <20 years in accordance with the World Health Organization definition [15]. Complete blood counts (CBC), C-reactive protein levels (CRP), and blood cultures were routinely taken for all OHD neonates. Cerebrospinal fluid (CSF) and gastric juice (GJ) cultures were collected when patients exhibited signs or symptoms of sepsis, including fever, respiratory distress with desaturation, seizure, apnea, tachycardia or bradycardia [16, 17].

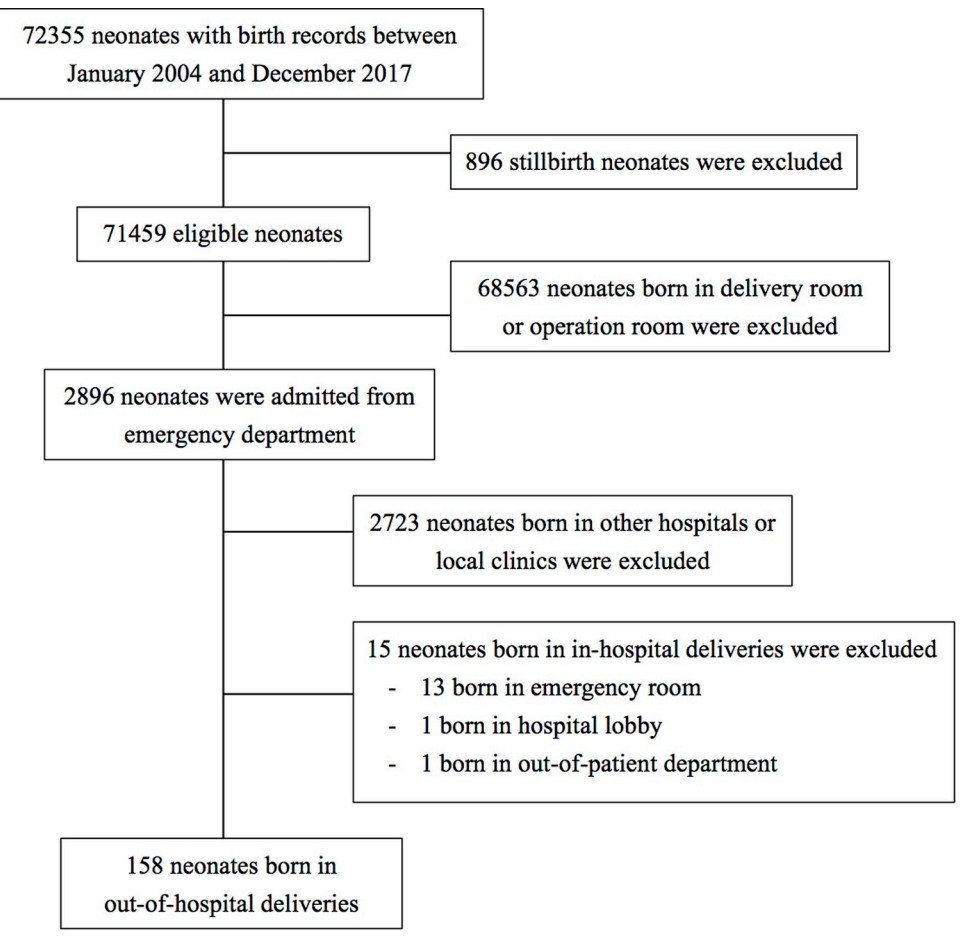

**Fig 1. Flowchart of the included neonates born in out-of-hospital deliveries.**

## Outcome measures

The primary outcome was the infection rate in the OHD group in comparison with that of the in-hospital deliveries. The criteria for neonatal infection were (1) positive culture from a sterile site, such as blood or CSF, and (2) positive GJ culture accompanied by clinical signs of sepsis. We limited the study to pathogens proven within 72 hours after birth since neonatal infection occurring in the first 72 hours of life is typically caused by organisms transmitted vertically from the mother before or during delivery [17–19], indicating that the pathogens are related to early-onset infections. In the OHD group, factors associated with infection were investigated by dividing the OHD neonates into infection group and non-infection group. We also analyzed the effect of gender, birth weight, GA, places of birth, suboptimal cord practices, and lack of antenatal care.

## Statistical analysis

Continuous variables were expressed as means ± standard deviations and compared using Student's *t* test. Categorical variables were expressed as numbers and percentages and compared using a chi-square or Fisher's exact test. A *P*-value of < 0.05 was considered statistically significant. All probabilities were two tailed. Odds ratios (ORs) with 95% confidence intervals (CIs) were calculated. Variables with a *P*-value of < 0.2 in the univariate analysis were included in

the multivariate logistic regression models. All analyses were performed using Microsoft Excel version 14.6.4 (Redmond, WA, USA) and IBM SPSS Statistics for Macintosh version 23.0 (Armonk, NY, USA).

## Results

### Characteristics of OHD neonates

A total of 158 OHD neonates were identified; all were singletons. The study accounted for 0.22% of the 71,459 live births during the study period. The male:female ratio was 0.98, and the GA ranged from 27 to 41 weeks with a mean age of 37.5 ± 2.8 weeks. Among the 125 neonates with an obtainable GA, 29 (23.2%) were preterm; this rate of preterm delivery was higher than the 12.9% in the control group. The mean birth weight of the OHD neonates was 2768.7 ± 549.8 g, and 43 (27.2%) of them had LBW. Their median hospital stay was 7 days, and 25 neonates (15.8%) were admitted to the ICU. A total of eight neonates (5.1%) received cardiopulmonary resuscitation (CPR), and 10 (6.3%) required endotracheal intubation. No mortality cases were recorded.

### Comparison of neonates with OHDs and in-hospital deliveries

Table 1 displays the characteristics of the neonates from OHDs and those from in-hospital deliveries. The OHD neonates had significantly younger mothers, higher rates of prematurity and LBW, and an increased risk of infection (5.1% vs. 0.5%; OR, 11.00; 95% CI, 5.36–22.58; $P < 0.001$). No significant difference in gender was noted. The maternal age in the OHD group ranged from 14 to 44 years with a mean of 29.4 ± 6.8 years. Twenty mothers (12.7%) were teenagers, 109 mothers (69%) were multiparous, 49 (31%) did not receive any prenatal care, and 10 (6.3%) claimed to be unaware of their pregnancies. Fig 2 shows the distribution of maternal age. Most mothers in both the OHD and in-hospital birth groups were between 20 and 39 years old, which is the usual childbearing age. The percentage of OHD mothers younger than 20 years (12.7%) was higher than that among the mothers without OHDs (1.2%; OR, 18.80; 95% CI, 11.15–30.61; $P < 0.001$), indicating that teenage mothers had an increased risk of OHD.

### Risk of infection in OHD neonates

In total, eight (5.1%) of the 158 OHD neonates had culture-confirmed infection by the age of 72 hours. As shown in Table 2, the neonates with infection had a younger GA, higher rate of prematurity and LBW, longer duration of antibiotic use, and a more complicated clinical course than did those without infection, as confirmed by the univariate analysis. Three

**Table 1. The characteristics and infection rates of neonates in out-of-hospital deliveries and those in in-hospital deliveries.**

| Variable | OHDs | In-hospital deliveries | OHDs | *P*-value |
|---|---|---|---|---|
| | (N = 158) | (N = 71301) | OR (95% CI) | |
| Maternal mean age ± SD (years) | 29.4 ± 6.8 | 32.1 ± 4.6 | Not applicable | <0.001 |
| Male:female ratio | 0.98 | 1.08 | 0.9 (0.66–1.23) | 0.524 |
| Teenage mother | 20 (12.7%) | 886 (1.2%) | 18.80 (11.15–30.61) | <0.001 |
| Prematurity | 29 (23.2%) | 9179 (12.9%) | 2.04 (1.35–3.10) | <0.001 |
| Low birth weight (<2500)(g) | 43 (27.2%) | 7409 (10.4%) | 3.22 (2.27–4.58) | <0.001 |
| Neonatal infection | 8 (5.1%) | 344 (0.5%) | 11.00 (5.36–22.58) | <0.001 |

OHDs, out-of-hospital deliveries; SD, standard deviation; OR, odds ratio; CI, confidence interval.

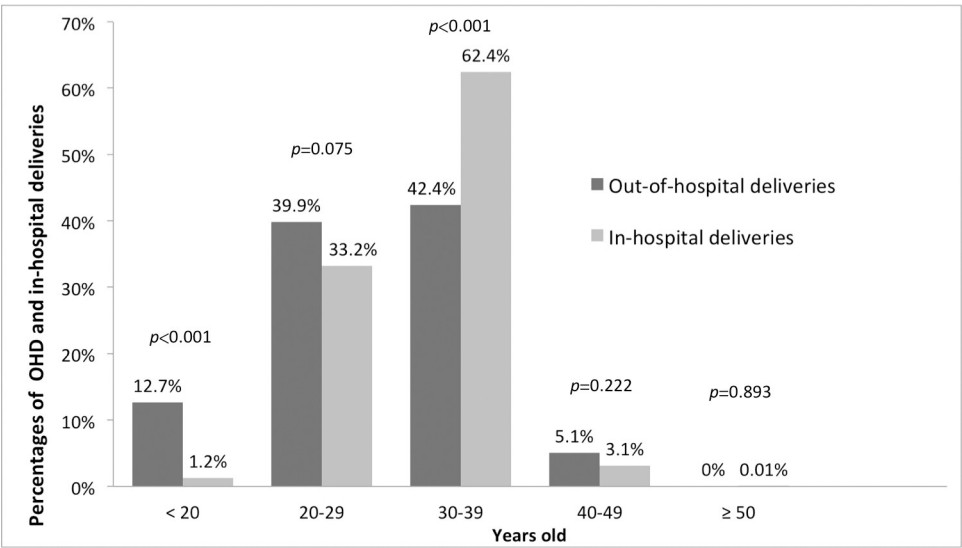

**Fig 2. Maternal age for neonates born in out-of-hospital deliveries and in-hospital deliveries.**

neonates in the infection group and 30 neonates in the non-infection group had unknown GA. No significant differences in gender, birth weight, use of antibiotics, suboptimal umbilical cord practices, CBC values, or CRP levels were observed in the OHD neonates with and without infection. Primipara and lack of prenatal care were more noticeable in the neonates with infection. In the multivariate analysis of the OHD neonates, LBW (OR, 0.16; 95% CI, 0.03–0.96; $P = 0.044$) was the only independent factor of neonatal infection.

The most frequent place of birth was at home (76, 48.1%), followed by in toilet bowls (22, 13.9%), in ambulances (16, 10.1%), in bathrooms (15, 9.5%), in private cars (14, 8.9%), in taxis (9, 5.7%), on sidewalks (3, 1.9%), and in other places (3, 1.9%). The places of birth in the infection and the non-infection groups were not significantly different.

### Etiology in OHD neonates with infection

Table 3 displays the characteristics and etiologies of the eight neonates with infection. Neonatal infections were confirmed by positive blood cultures in one neonate (*Enterococcus faecium*) and GJ cultures in seven neonates. The most common pathogens were *Escherichia coli* (three GJ isolates) and *Klebsiella spp*. (three GJ isolates). Twelve OHD neonates were performed spinal tapping, and all the CSF cultures were negative findings.

In OHD neonates, 152 (96.2%) neonates were administered antibiotics because of signs or symptoms of infection after birth, contaminated places of birth or suboptimal cord cutting. Ninety-seven patients (61.4%) received ampicillin and gentamicin, 27 patients (17.1%) received penicillin and gentamicin, and 11 (7.0%) received oxacillin and gentamicin. The mean duration of antibiotic use was 4.2 ± 2.2 days. Twenty-nine OHD neonates (22.7%) experienced suboptimal umbilical cord separation. One neonate with a congenital anomaly and unsterile cord cutting was administered prophylactic tetanus immune globulin.

### Discussion

In this study, neonatal infection in OHDs was 11-fold higher than that in in-hospital deliveries, and LBW was the only factor independently associated with infection risk in OHDs. The infection rate of OHD neonates has not been previously documented. Contaminated places of

**Table 2. Univariate analysis of variables in neonates born outside hospitals with and without infection in the first 72 hours of life.**

| Variables | Infection (N = 8) | Non-infection (N = 150) | OR (95% CI) | *P*-value |
|---|---|---|---|---|
| Gestational age (weeks) | 33.3 ± 3.5 | 37.8 ± 2.7 | NA (-8.83– -0.17) | 0.045 |
| Prematurity | 4 (80.0%) | 25 (21.0%) | 3.95 (0.68–22.85) | 0.011 |
| Unknown gestational age | 3 (37.5%) | 30 (20%) | 2.30(0.54–10.61) | 0.235 |
| Male | 4 (50.0%) | 74 (49.3%) | 1.03 (0.25–4.26) | 0.971 |
| Birth weight (g) | 2224.6 ± 712.9 | 2797.8 ± 527.2 | NA (-1171.06–24.79) | 0.058 |
| Low birth weight (<2500)(g) | 6 (75.0%) | 37 (24.7%) | 9.16 (1.77–47.37) | 0.005 |
| Maternal age (years) | 30.1 ± 8.6 | 29.4 ± 6.8 | NA (-6.49–7.98) | 0.818 |
| Teenage mother | 2 (25%) | 18 (12%) | 2.00 (0.22–17.89) | 0.281 |
| Lack of antenatal care | 5 (62.5%) | 44 (29.3%) | 4.00 (0.92–17.53) | 0.048 |
| Primipara | 5 (62.5%) | 44 (29.3%) | 4.00 (0.92–17.53) | 0.048 |
| Suboptimal cord practices | 2 (40%) | 27 (22.0%) | 2.37 (0.38–14.92) | 0.317 |
| ICU | 6 (75.0%) | 19 (12.7%) | 20.68 (3.89–109.99) | <0.001 |
| CPR | 2 (25.0%) | 6 (4.0%) | 8.00 (1.33–48.24) | 0.054 |
| Endotracheal intubation | 3 (37.5%) | 7 (4.7%) | 12.26 (2.43–61.94) | 0.009 |
| Use of antibiotics | 8 (100%) | 144 (96.0%) | 0.96 (0.93–0.99) | 0.564 |
| Days of antibiotic use | 7.0 ± 2.6 | 4.1± 2.1 | NA (0.71–5.10) | 0.016 |
| Toilet or bathroom | 4 (50%) | 33 (22.0%) | 3.55 (0.84–14.95) | 0.088 |
| Laboratory data | | | | |
| Hemoglobin (g/dL) | 18.9 ± 2.2 | 18.6 ± 3.8 | NA (-1.48–2.28) | 0.643 |
| WBC count (/μL) | 24425 ± 21929 | 16528 ± 5866 | NA (-10444.16–26237.21) | 0.343 |
| Platelet ($10^3$/μL) | 230.6 ± 66.2 | 286.2 ± 79.7 | NA (-111.38–0.31) | 0.051 |
| CRP (mg/dL) | 0.4 ± 0.9 | 0.1 ± 0.6 | NA (-0.51–0.95) | 0.494 |

ICU, intensive care unit; CPR, cardiopulmonary resuscitation; WBC, white blood cells; CRP, C-reactive protein; OR, odds ratio; CI, confidence interval; NA, not applicable.

birth, suboptimal cord cutting and poor perinatal care may be associated with the infection risk of neonates. The umbilical cord can serve as an entry point for bacteria [20, 21]. Pathogens can directly access the bloodstream via the patent vessels of the newly cut cord. Such infections are preventable and can be reduced by hygienic delivery and sterile cord care. However,

**Table 3. Characteristics of neonates born outside hospitals with infection within 72 hours after birth.**

| Case | GA (weeks) | BW (g) | Maternal age (years) | Place of birth | Culture |
|---|---|---|---|---|---|
| 1 | 37[+4] | 2890 | 28 | Ambulance | *Escherichia coli* (GJ), *Klebsiella pneumoniae* (GJ), *Staphylococcus aureus* (MSSA) (GJ) |
| 2 | 30 | 2076 | 18 | Toilet bowl | *Escherichia coli* (GJ), *Klebsiella oxytoca* (GJ), *Morganella morganii* (GJ), *Viridans streptococci* (GJ) |
| 3 | 36[+3] | 2350 | 35 | Home | *Enterococcus faecium* (blood) |
| 4 | unknown | 3600 | 34 | Home | fungi (GJ) |
| 5 | unknown | 2032 | 16 | Toilet bowl | *Klebsiella oxytoca* (GJ) |
| 6 | 31[+6] | 1704 | 38 | Home | *Neisseria gonorrhoeae* (GJ) |
| 7 | 30[+3] | 1690 | 36 | Toilet bowl | *Candida albicans* (GJ) |
| 8 | unknown | 1455 | 36 | Bathroom | *Escherichia coli* (GJ) |

GA, gestational age; BW, birth weight; GJ, gastric juice; MSSA, methicillin-susceptible *Staphylococcus aureus*.

**Table 4. Comparative variables of neonates born in out-of-hospital deliveries in other studies.**

| First author | Country | Interval | Cases | Incidence | Maternal age (years) | Multiparous mother | Lack of antenatal care | GA (weeks) | BW (g) | Preterm | ICU |
|---|---|---|---|---|---|---|---|---|---|---|---|
| This study | Taiwan | 2004–2017 | 158 | 0.22% | 29.4 ± 6.8 (14–44) | 69% | 31% | 37.5 ± 2.8 (27–41) | 2768 | 23.2% | 15.8% |
| McLelland G | Australia | 2000–2010 | 313 | 0.45% | 29.9 ± 5.8 (16–44) | 90.5% | 3.7% | 38.4 ± 3.6 (20–42) | | 11% | |
| Rodie VA | United Kingdom | 1995–1999 | 121 | 0.6% | 26 (15–44) | 88.7% | 24.3% | 39.3 (23–42) | 3000 | | 54.3% |
| Lazic Z | Slovenia | 1997–2005 | 58 | 0.32% | | 79.3% | 30% | | | 22% | |
| Unterscheider J | Ireland | 2005–2009 | 143 | 0.36% | 30 (18–43) | 92.3% | | 38.4 | 3138 | 12.5% | 8.1% |
| Ramsewak S | West Indies | 1987–1993 | 326 | 0.81% | | | 29.7% | | | | |
| L Renesme | France | 2007–2009 | 76 | 0.42% | 30 (16–41) | 90.7% | 27.4% | 40 (25–42) | 3130 | 7.4% | 14.5% |
| Katja Ovaskainen | Finland | 1996–2011 | 67 | 0.1% | 29 (15–47) | | 12% | 39.7 | 3460 | | 19% |
| François Javaudin | France | 2011–2018 | 1670 | <1% | 30 ± 5.5 (15–48) | 87% | 6.5% | 38 | 3008 | 8.1% | 6.3% |
| Sikhulile Khupakonke | South Africa | 2015–2016 | 201 | 4.6% | 27 | 89.8% | 16.7% | | 2689 | 35.2% | 6.4% |

GA, gestational age; BW, birth weight; ICU, intensive care unit.

suboptimal cord practices did not contribute to the risk of infection in this study, possibly because of cleaning of the cord stump soon after hospitalization as well as antibiotic administration [22]. The occurrence of infection did not differ among places of birth, even for toilet bowls or bathrooms, which are generally considered to be unhygienic. The practice of antibiotic administration in the majority of the OHD neonates may have reduced the incidence of infection and conceal the adverse factors of the place where labor occurred. On account of the characteristics of poor prenatal care among the OHD mothers, missing information on GA (20.9%) was common in our study. As birth weight and GA were confounding factors, LBW may more precisely represent factors associated with infection in OHD neonates instead of GA. In addition, preterm neonates and neonates with LBW are at increased risk of infection, neonatal morbidity and mortality compared with normal-weight full-term infants [23, 24]. Consequently, complete postnatal care and subsequent follow-up are crucial for OHD neonates, especially those with LBW.

Over the study period, the incidence of OHD in this study was 0.22%, which is similar to that reported in other studies (Table 4), and our findings were in line with previous studies of accidental OHDs [1–8, 11]. OHDs have been reported to be associated with the risk of prematurity [1–3, 25]. In our study, prematurity was significantly higher for the OHDs than for the in-hospital births. Compared with in-hospital deliveries, previous studies have indicated that ICU admission is more prevalent for OHDs, indicating a more complicated clinical course [1, 2, 4–6]. Teenage mothers and preterm deliveries were found to pose a risk of OHDs. The soft birth canal in young women and small preterm infants may cause rapid delivery [25]. Multiparous mothers and lack of prenatal care were also related to OHDs in previous reports [1–5, 9]. Furthermore, OHDs may cause high neonatal mortality and morbidity, especially in

developing countries and areas short of medical assistance [11]. Prompt intervention and care may be required for all OHD neonates to improve outcomes.

A consensus has not yet been reached on empirical antibiotic use and the pathogens associated with OHDs. The antibiotic combination of ampicillin/penicillin/oxacillin and gentamicin were used for the majority of the OHD neonates in the present study. We attempted to identify pathogens related to OHDs by collecting blood cultures, CSF and GJ cultures before antibiotic use. Several studies have proposed that gastric aspirates are related to amniotic fluid leaks and neonatal infection [26–29]. The neonatal gastrointestinal tract is considered sterile, but diverse microbiota flora is present soon after birth [30, 31]. Preterm neonates are more susceptible to colonization by potentially pathogenic bacteria, resulting in an increased risk of necrotizing enterocolitis [32–34]. The mechanism of the gastric pathogens in neonatal infection was inconclusive. The isolation of pathogens from neonatal gastric contents has been reported to be a source of systemic infection [35]. Consequently, gastric aspirates may be helpful for detecting bacterial infection, especially when neonates exhibit signs of sepsis. Consistent with our study, Stewart et al. and Sawardekar et al. identified *Escherichia coli*, *Klebsiella spp*., and *Staphylococcus aureus* as the most common pathogens isolated in umbilical cord cultures in infants born at home with omphalitis [20, 36]. Although no umbilical cultures were collected in our study, the pathogens isolated from the blood and GJ cultures were similar to those in the OHD infants with omphalitis. Such pathogens might be considered if infection is suspected in clinical practice.

### Limitations

Limitations of this study included the unavailability of Apgar scores due to OHDs, which made birth status unclear. Second, in this retrospective study, confounding factors were a concern. Several of the variables were not independent of each other, such as GA and birth weight. In the statistical analysis, we attempted to adjust the confounding factors and predict the risk factors in accidental OHDs. Third, the sample size of the OHDs was relatively small, and neonatal infection related to OHDs has not been investigated, as evidenced in previous studies [1–8, 11]. Therefore, additional studies with more cases are required to elucidate the infection risk of OHDs. Finally, empirical antibiotics were administered to the majority of the neonates in our study, meaning the true risk of infection in OHD neonates may have been underestimated. A prospective study can clarify the effectiveness of empirical antibiotics in these neonates and determine the OHD-related pathogens.

### Conclusions

We demonstrated that the infection rate was higher for OHDs than for in-hospital deliveries, and infection rate was associated with LBW. Inpatient care might be needed for accidental OHD neonates. Because poor antenatal care, prematurity, and teenage pregnancy were relatively common in the OHDs, social support services should be provided for vulnerable populations, such as teenage mothers.

### Supporting information

**S1 Data.**
(XLSX)

### Author Contributions

**Conceptualization:** Chia-Jung Chang, Lung Chang.

**Data curation:** Chia-Jung Chang, Hsin Chi, Lung Chang.

**Formal analysis:** Chia-Jung Chang, Hsin Chi, Wai-Tim Jim, Lung Chang.

**Investigation:** Wai-Tim Jim.

**Methodology:** Chia-Jung Chang, Hsin Chi, Wai-Tim Jim, Nan-Chang Chiu, Lung Chang.

**Resources:** Chia-Jung Chang.

**Software:** Chia-Jung Chang, Hsin Chi.

**Supervision:** Chia-Jung Chang, Hsin Chi, Wai-Tim Jim, Nan-Chang Chiu, Lung Chang.

**Writing – original draft:** Chia-Jung Chang, Lung Chang.

**Writing – review & editing:** Chia-Jung Chang, Hsin Chi, Wai-Tim Jim, Nan-Chang Chiu, Lung Chang.

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
