## [Decision Letter · Decision Letter 0]

11 Aug 2021

PONE-D-21-20144

Risk of infection in neonates born in accidental out-of-hospital deliveries.

PLOS ONE

Dear Dr. Lung Chang

Thank you for submitting your manuscript to PLOS ONE. After careful consideration, we feel that it has merit but does not fully meet PLOS ONE’s publication criteria as it currently stands. Therefore, we invite you to submit a revised version of the manuscript that addresses the points raised during the review process.

Please attend to the concerns of the reviewers and re-submit should you wish to.

Please submit your revised manuscript by 25 September. If you will need more time than this to complete your revisions, please reply to this message or contact the journal office at plosone@plos.org. Please include the following items when submitting your revised manuscript:

We look forward to receiving your revised manuscript.

Kind regards,

Lloyd J. Tooke, MBChB, MMed(Paeds)

Academic Editor

PLOS ONE

2. In ethics statement in the manuscript and in the online submission form, please provide additional information about the patient records/samples used in your retrospective study. Specifically, please ensure that you have discussed whether all data/samples were fully anonymized before you accessed them and/or whether the IRB or ethics committee waived the requirement for informed consent. If patients provided informed written consent to have data/samples from their medical records used in research, please include this information.

Important: If there are ethical or legal restrictions to sharing your data publicly, please explain these restrictions in detail. Please see our guidelines for more information on what we consider unacceptable restrictions to publicly sharing data: http://journals.plos.org/plosone/s/data-availability#loc-unacceptable-data-access-restrictions. Note that it is not acceptable for the authors to be the sole named individuals responsible for ensuring data access

Reviewers' comments:

Reviewer's Responses to Questions

**Comments to the Author**

1. Is the manuscript technically sound, and do the data support the conclusions?

Reviewer #1: Yes

Reviewer #2: Yes

2. Has the statistical analysis been performed appropriately and rigorously? 

Reviewer #1: Yes

Reviewer #2: Yes

3. Have the authors made all data underlying the findings in their manuscript fully available?

Reviewer #1: Yes

Reviewer #2: Yes

4. Is the manuscript presented in an intelligible fashion and written in standard English?

Reviewer #1: Yes

Reviewer #2: No

5. Review Comments to the Author

Reviewer #1: Thank you for this excellent piece of work. Please could you clarify these issues:

1. Regarding the title, it would be valuable to state where the study took place, and whether it was in an urban, peri urban or rural setting. This helps those of us who have never been to your unit understand the context better.

2. Again, the place of study should be included in the abstract for clarity.

3. The outborn cohort is very small, but this certainly speaks to the excellent care provision in your setting. In your setting, are many babies born electively at home, under the care of a doula or midwife?

4. Please clarify this sentence (on line 57 - 58): " Duration and antimicrobial therapy were recorded."

5. Please define what the local antibiotic policy is, in reference to the statement on line 141 - 142: In accordance with the common practices of the facility, 152 (96.2%) OHD neonates were administered antibiotics".

6. This sentence on line 163 ". Khupakonke et al. also demonstrated LBW to be a predicting factor of OHD [11]." feels out of place, and should probably be kept in the introduction.

7. The rest of the paper does not mention the prognosis of the patients, so the sentence on line 167 "The prognostic outcomes of OHD neonates require further research. " does not need to be included.

Reviewer #2: Abstract:

Page 2 line 14, the sentence “the OHD group was comprised of more teenage mothers than adult mothers” technically is not right. The OHD group comprised of more teenage mothers compared to the in-hospital deliveries category.

Study variables:

Does Neonatal age here refer to Gestational age?

“Cerebrospinal fluid (CSF) and gastric juice (GJ) cultures were collected when patients exhibited signs or symptoms of sepsis, including fever, respiratory distress with desaturation, seizure, apnea, tachycardia or bradycardia” but there is no mention of CSF in the results.

Outcome measures:

Page 6 line 72: “…indicating that pathogens are related to OHD.” Granted, infections associated with OHDs are more likely to be Early Onset infections but sentence should be rephrased since it is ambiguous and gives an impression of causality.

Results:

Characteristics of OHD neonates - Hospital stay would be better expressed as a median

Table 2 – Some of the percentages are not correct both under the infection & non infection columns.

What denominators were used for the Prematurity calculations?

The sub-optimal cord practices percentages for both the infection and non-infection columns are incorrect, should be 25% & 18% respectively.

Hospital stay would be better expressed as a median

Etiology of OHD in neonates with infection:

The text on the various pathogens isolated from the cultures should be rewritten as it contains a fair amount of redundancy. Since there was only one blood isolate with the remainder as GJs, it can be made to reflect that as opposed to the numerous GJ mentions.

Page 12 line 145 – percentage value of neonates with sub-optimal cord separation is supposed to be 18.4% based on a denominator of 158, as opposed to the 22.7% that is reported

General comments:

Percentage values for Prematurity seems to be inconsistent throughout the study, sometimes it is 23.2% (Page 2 line 8, page 8 table 1 etc.) and at other times 18.4% (Page 15 Table 4). It would seem different denominators are used interchangeably? When is the 158 used and when is the 125 used?

Accidental, Unexpected and Unplanned seems to be used interchangeably to qualify OHDs. Can the authors stick to one? Perhaps “accidental” would be most appropriate since that is what is used in the title of the study.

Typographical / Grammatical errors

Page 2 line 6 /7 “This study also investigated risk factors…” should read “This study also investigated both maternal and neonatal risk factors…”

Page 2 line 13 should read “Forty-nine (31%) OHD mothers lacked antenatal care…”

Page 2 line 14 / 15 should read “Neonatal infection was more prevalent among…”

Page 16 line 187 should read “The neonatal gastrointestinal tract is considered sterile, but diverse microbiota flora is present soon after birth”

Page 16 line 188, “colonization” typo error

6. PLOS authors have the option to publish the peer review history of their article (what does this mean?). If published, this will include your full peer review and any attached files.

Reviewer #1: **Yes: **Lyndal Gibbs

Reviewer #2: No

---

## [Author Response · Author response to Decision Letter 0]

17 Oct 2021

Response to Reviewers

Reviewer #1:

1. Regarding the title, it would be valuable to state where the study took place, and whether it was in an urban, peri urban or rural setting. This helps those of us who have never been to your unit understand the context better.

→ Thanks for your suggestion. This study was conducted in an urban setting of the developed country, and we had added this information in the first paragraph of Materials and Methods. (Design and setting: Page 4, line 45-46)

2. Again, the place of study should be included in the abstract for clarity.

→ Thanks for your comments. We had added this information in the abstract. (Abstract: Page 2, line 4-6)

3. The outborn cohort is very small, but this certainly speaks to the excellent care provision in your setting. In your setting, are many babies born electively at home, under the care of a doula or midwife?

→ During the 14 years of study period, none of the neonates was sent to our hospital after electively born at home. Perhaps due to the sound and prevalent National Health Insurance in Taiwan, most pregnant women prefer delivery at medical facilities for the sake of more medical assistance with mothers and neonates, so planned home birth is not common in our country.

4. Please clarify this sentence (on line 57 - 58): " Duration and antimicrobial therapy were recorded."

→ Thanks for your comments. We had added the clarification: We recorded the duration of antimicrobial therapy in the first 72 hours of neonatal life for early-onset infections.

(Materials and Methods: Study variables, Page 5, line 60-62)

5. Please define what the local antibiotic policy is, in reference to the statement on line 141 - 142: In accordance with the common practices of the facility, 152 (96.2%) OHD neonates were administered antibiotics".

→ Thanks for your kind remarks. It is my mistake of imprecise grammar. I had revised as the following: In OHD neonates, 152 (96.2%) neonates were administered antibiotics because of signs or symptoms of infection after birth, contaminated places of birth or suboptimal cord cutting.

 (Results: Etiology in OHD neonates with infection, Page 12, line 145-146)

6. This sentence on line 163 ". Khupakonke et al. also demonstrated LBW to be a predicting factor of OHD [11]." feels out of place, and should probably be kept in the introduction.

→ Thanks for your suggestions. I had shifted this sentence to the part of introduction. (Introduction: Page 3, line 30-31)

7. The rest of the paper does not mention the prognosis of the patients, so the sentence on line 167 "The prognostic outcomes of OHD neonates require further research. " does not need to be included.

→ Thanks for your comments. I had deleted this sentence.

Reviewer #2: Abstract:

Page 2 line 14, the sentence “the OHD group was comprised of more teenage mothers than adult mothers” technically is not right. The OHD group comprised of more teenage mothers compared to the in-hospital deliveries category.

→ Thank you. I had revised as your suggestions. (Abstract: Page 2, line 14-15)

Study variables:

Does Neonatal age here refer to Gestational age?

→ Yes. I had corrected neonatal age to gestational age. 

(Materials and Methods: Study variables, Page 5, line 58)

“Cerebrospinal fluid (CSF) and gastric juice (GJ) cultures were collected when patients exhibited signs or symptoms of sepsis, including fever, respiratory distress with desaturation, seizure, apnea, tachycardia or bradycardia” but there is no mention of CSF in the results.

→ Thanks for your comments. Twelve OHD neonates were received spinal tapping, and all CSF cultures were negative findings. I had added in the part of results. 

(Results: Etiology in OHD neonates with infection, Page 11, line 138-139)

Outcome measures:

Page 6 line 72: “…indicating that pathogens are related to OHD.” Granted, infections associated with OHDs are more likely to be Early Onset infections but sentence should be rephrased since it is ambiguous and gives an impression of causality.

→ Thanks for your kind remarks. We had made the revisions. 

We limited the study to pathogens proven within 72 hours after birth since neonatal infection occurring in the first 72 hours of life is typically caused by organisms transmitted vertically from the mother before or during delivery, indicating that the pathogens are related to early-onset infections. 

(Materials and Methods: Outcome measures, Page 6, line 73-76)

Results:

Characteristics of OHD neonates - Hospital stay would be better expressed as a median

→ Thanks for your comments. We changed the mean hospital stay to the median hospital stay which was 7 days.

(Results: Characteristics of OHD neonates, Page 7, line 96)

Table 2 – Some of the percentages are not correct both under the infection & non infection columns.

What denominators were used for the Prematurity calculations?

→ Since the characteristics of poor prenatal care among the OHD mothers, 33 neonates with unknown gestational age (GA) were deducted from denominators. Three neonates in the infection group and 30 neonates in the non-infection group had unknown GA. In the infection group (N=8), 5 neonates with obtainable GA, 4 (4/5, 80%) were preterm. In the non-infection group (N=150), 120 neonates with obtainable GA, 25 (25/120, 21%) were preterm. 

→ We had added the variable of unknown GA in Table 2 and illustrations in line 120-121 to avoid misleading. 

(Results: Risk of infection in OHD neonates, Page 9, line 120-121)

The sub-optimal cord practices percentages for both the infection and non-infection columns are incorrect, should be 25% & 18% respectively.

→ Thirty neonates with missing information of cord practices were deducted from denominators. Three neonates and 27 neonates had unknown methods of cord practices in the infection group and non-infection group, respectively. In the infection group (N=8), 2 (2/5, 40%) neonates had suboptimal cord practices. In the non-infection group (N=150), 27 (27/123, 22%) neonates received suboptimal cord practices.

Hospital stay would be better expressed as a median

→ We had shifted to the median hospital stay in text. (Results: Characteristics of OHD neonates, Page 7, line 96)

Etiology of OHD in neonates with infection:

The text on the various pathogens isolated from the cultures should be rewritten as it contains a fair amount of redundancy. Since there was only one blood isolate with the remainder as GJs, it can be made to reflect that as opposed to the numerous GJ mentions.

Thanks for your suggestions. I had rewritten this paragraph. 

(Results: Etiology in OHD neonates with infection, Page 11, line 134-139)

Page 12 line 145 – percentage value of neonates with sub-optimal cord separation is supposed to be 18.4% based on a denominator of 158, as opposed to the 22.7% that is reported

→ Since 30 neonates with missing information of cord practices had deducted from denominators, percentage value of neonates with sub-optimal cord separation was 22.7% (29/128).

General comments:

Percentage values for Prematurity seems to be inconsistent throughout the study, sometimes it is 23.2% (Page 2 line 8, page 8 table 1 etc.) and at other times 18.4% (Page 15 Table 4). It would seem different denominators are used interchangeably? When is the 158 used and when is the 125 used?

→ 33 neonates with unknown GA were deducted from denominators. In the OHD group, 29 neonates were preterm, so the percentage value for prematurity was 23.2% (29/125).

→ Thanks for your kind remarks. It is my mistake of wrong taping of percentage of prematurity in Page 15 Table 4. I had revised the percentage values for prematurity to 23.2% in Table 4. (Discussion: Page 15, Table 4)

Accidental, Unexpected and Unplanned seems to be used interchangeably to qualify OHDs. Can the authors stick to one? Perhaps “accidental” would be most appropriate since that is what is used in the title of the study.

→ Thanks for your suggestions. We had rewritten the manuscript with consistency of“accidental”.

Typographical / Grammatical errors

Page 2 line 6 /7 “This study also investigated risk factors…” should read “This study also investigated both maternal and neonatal risk factors…”

→ Thanks. We had made the revision. (Abstract: Page 2, line 7)

Page 2 line 13 should read “Forty-nine (31%) OHD mothers lacked antenatal care…”

→ We had corrected this error. (Abstract: Page 2, line 13-14)

Page 2 line 14 / 15 should read “Neonatal infection was more prevalent among…”

→ We had revised as your suggestions. (Abstract: Page 2, line 15-16)

Page 16 line 187 should read “The neonatal gastrointestinal tract is considered sterile, but diverse microbiota flora is present soon after birth”

→ We had made the revision. (Discussion: Page 16, line 191-192)

Page 16 line 188, “colonization” typo error

→ Thanks for your kind remarks. We had corrected this error. (Discussion: Page 16, line 192)

---

## [Decision Letter · Decision Letter 1]

28 Jan 2022

Risk of infection in neonates born in accidental out-of-hospital deliveries.

PONE-D-21-20144R1

Dear Dr. Chang,

We’re pleased to inform you that your manuscript has been judged scientifically suitable for publication and will be formally accepted for publication once it meets all outstanding technical requirements.

Kind regards,

Kazumichi Fujioka

Academic Editor

PLOS ONE

Additional Editor Comments (optional):

Reviewers' comments:

Reviewer's Responses to Questions

**Comments to the Author**

1. If the authors have adequately addressed your comments raised in a previous round of review and you feel that this manuscript is now acceptable for publication, you may indicate that here to bypass the “Comments to the Author” section, enter your conflict of interest statement in the “Confidential to Editor” section, and submit your "Accept" recommendation.

Reviewer #2: (No Response)

Reviewer #3: All comments have been addressed

2. Is the manuscript technically sound, and do the data support the conclusions?

Reviewer #2: Yes

Reviewer #3: Yes

3. Has the statistical analysis been performed appropriately and rigorously? 

Reviewer #2: Yes

Reviewer #3: Yes

4. Have the authors made all data underlying the findings in their manuscript fully available?

Reviewer #2: Yes

Reviewer #3: Yes

5. Is the manuscript presented in an intelligible fashion and written in standard English?

Reviewer #2: Yes

Reviewer #3: Yes

6. Review Comments to the Author

Reviewer #2: Abstract

Line 5: Could it be specified where the urban setting is? For example, urban setting in Taiwan?

Statistical analyses

Line 81: Since length of hospital stay has now been expressed as a median, the statistical analyses should be modified to reflect that.

Typographical errors

Line 16: Neonatal infection was more prevalent among OHDs than for in hospital deliveries

Line 25: OHD should be made plural

Line 93: The GA ranged from 27 to 41 weeks with a mean of …. (The word “age” should be deleted)

Line 105: … and 10 (6.3%) claimed to be unaware of their pregnancy. The word “claimed” here seems inappropriate, gives the impression, the authors do not believe the mothers. Merriam-Webster defines the word claim as “to assert in the face of possible contradiction”. Is that the case here?

Line 130: should read “inside toilet bowls” instead of ‘in toilet bowls’

Line 132: should read “other places” instead of ‘in other places’

Line 138: Twelve OHD neonates had a spinal tap done and all CSF cultures were sterile.

Line 164: …conceal the adverse “effects” of the place where labour occurred, instead of conceal the adverse “factors” of the place where labour occurred

Reviewer #3: Review comments on Manuscript Number: ONE-D-21-20144R1. Entitled "Risk of infection in neonates born in accidental out-of-hospital deliveries."

Overall, the idea of research is very interesting, organized and well written reasonable. The authors have done great effort to accomplish this work. They fulfilled all reviewers' comments and made necessary changes throughput the manuscript.

7. PLOS authors have the option to publish the peer review history of their article (what does this mean?). If published, this will include your full peer review and any attached files.

Reviewer #2: **Yes: **Dr Naana Ayiwa Wireko Brobby

Reviewer #3: No

---

## [Editor Report · Acceptance letter]

2 Feb 2022

PONE-D-21-20144R1 

Risk of infection in neonates born in accidental out-of-hospital deliveries 

Dear Dr. Chang:

I'm pleased to inform you that your manuscript has been deemed suitable for publication in PLOS ONE. Congratulations! Your manuscript is now with our production department. 

Kind regards, 

on behalf of

Dr. Kazumichi Fujioka 

Academic Editor

PLOS ONE